# The Role of Neuropilin-2 in the Epithelial to Mesenchymal Transition of Colorectal Cancer: A Systematic Review

**DOI:** 10.3390/biomedicines10010172

**Published:** 2022-01-14

**Authors:** Cristina Lungulescu, Valentina Ghimpau, Dan Ionut Gheonea, Daniel Sur, Cristian Virgil Lungulescu

**Affiliations:** 1Doctoral School, University of Medicine and Pharmacy Craiova, 2 Petru Rares Str., 200349 Craiova, Romania; cristina.lungulescu@yahoo.com (C.L.); valentinaghimpau@gmail.com (V.G.); 2Department of Gastroenterology, University of Medicine and Pharmacy Craiova, 2 Petru Rares Str., 200349 Craiova, Romania; dan.gheonea@umfcv.ro; 311th Department of Medical Oncology, University of Medicine and Pharmacy “Iuliu Hatieganu”, 400012 Cluj-Napoca, Romania; 4Department of Oncology, University of Medicine and Pharmacy Craiova, 2 Petru Rares Str., 200349 Craiova, Romania; cristian.lungulescu@umfcv.ro

**Keywords:** neuropilin-2, cancer, epithelial–mesenchymal transition, NRP2, metastasis, angiogenesis

## Abstract

Neuropilin-2 (NRP-2) expression has been found in various investigations on the expression and function of NRP-2 in colorectal cancer. The link between NRP-2 and colorectal cancer, as well as the mechanism that regulates it, is still mostly unclear. This systematic review was carried out according to the Cochrane guidelines for systematic reviews. We searched PubMed, Embase^®^, MEDLINE, Allied & Complementary Medicine^TM^, Medical Toxicology & Environmental Health, DH-DATA: Health Administration for articles published before 1 October 2021. The following search terms were used: “neuropilin-2” “neuropilin 2”, “NRP2” and “NRP-2”, “colorectal cancer”, “colon cancer”. Ten articles researching either tumor tissue samples, cell lines, or mice models were included in this review. The majority of human primary and metastatic colon cancer cell lines expressed NRP-2 compared to the normal colonic mucosa. NRPs have been discovered in human cancers as well as neovasculature. The presence of NRP-2 appears to be connected to the epithelial–mesenchymal transition’s function in cancer dissemination and metastatic evolution. The studies were heterogeneous, but the data assessed indicates NRP-2 might have an impact on the metastatic potential of colorectal cancer cells. Nevertheless, further research is needed.

## 1. Introduction

The process of epithelial to mesenchymal transition (EMT) is characterized by the acquisition of mesenchymal features by epithelial cells as a reaction to normal or pathological stimuli. As cells undergo EMT they lose their apical–basal polarity, suffer dissolution of the tight junction, and reconfigure their cytoskeletal architecture, allowing them to invade surrounding structures (Figure 1).

EMT is classified into three types, as it occurs throughout embryonic development and organ creation [1], tissue repair and fibrosis [2], and cancer progression [3]. Effector molecules carry out the EMT process, which is controlled by transcription factors—EMT core regulators—and is activated by external stimuli—EMT inducers [4], EMT is defined by the downregulation of genes encoding cell junction proteins (E-cadherin, claudins, and occludins) and the activation of genes expressing mesenchymal adhesion-promoting proteins (vimentin, fibronectin, and N-cadherin). EMT effectors are mostly subcellular structural proteins that influence the phenotypic of the cell. EMT core regulators are transcription factors that govern epithelial and mesenchymal markers by regulating target genes or adjusting their expression. The zinc-finger transcription factors SNAI1 and SNAI2, the basic helix–loop–helix factors TWIST1 and TWIST2, and the zinc-finger E-box–binding homeobox factors ZEB1 and ZEB2 are all significant EMT transcription factors [5].

Furthermore, numerous elements have been implicated as possible inducers of EMT in colorectal cancer. These inducers modify EMT-transcriptional factors, causing intracellular signaling cascades [5,6]. Among them, neuropilins (NRPs) are non-tyrosine kinase glycoproteins initially discovered in the nervous system. The neuropilin (NRP) family is comprised of two genes, neuropilin-1 (NRP-1) and neuropilin-2 (NRP-2), and participate in a variety of signaling pathways that contribute to the cytoskeletal structure, angiogenesis, and tumor development [7].

NRPs were shown to be expressed by a wide range of malignancies, indicating that this glycoprotein may have a role in cancer growth. NRP-2 expression has been found in colon cancers, pancreatic cancers, breast cancers, osteosarcomas, melanoma, lung cancers, brain tumors, myeloid leukemia, infantile hemangioma, salivary adenoid cystic carcinoma, and ovarian neoplasms [8,9,10,11,12,13,14,15,16,17,18,19,20,21,22,23,24]. Increasing VEGFR1 phosphorylation and activating the VEGFR1/Pi3K/Akt pathway, it was hypothesized, underlie the carcinogenic characteristics of NRP-2. However, the particular biochemical pathways triggered by NRP-2 and implicated in oncogenesis are still mainly unclear at this time. Endothelial growth factor (VEGF) and vascular semaphorin (SEMA) can bind to it, making it a receptor [25,26]. Moreover, lymphangiogenesis regulator NRP-2 has a receptor on both endothelial and cancerous cells, making it a key player in the process [27,28,29]. NRP-2 has been shown to grant cancer cells a fibroblastic morphology, which suggests its involvement in the process of EMT [5]. Furthermore, we illustrate the regulatory mechanism of NRP2 in CRC (Figure 2).

Either NRP-1 or NRP-2 or both are expressed in nearly all tumor cells. Tumor development, progression, and therapy options have all been examined for NRP-1 [30]. Solid tumor development is aided when NRP-1 binds VEGF-A, while SEMA3A binding often improves prognosis by limiting tumor cell motility and invasion. Furthermore, via activating the TGF-β pathway, NRP-1 promotes EMT and contributes to metastasis.

NRP-2’s connection to colorectal cancer is still under investigation, as is the method by which it is regulated. Research on the involvement of NRP-2 and EMT in colorectal cancer development is being systematically reviewed in the current paper.

This systematic review aims to highlight the ability of neuropilin-2 in the epithelial-mesenchymal transition in colorectal cancer cells. The key objective is to provide an up-to-date literature review on the role of NRP-2 correlated to EMT, particularly in the progression of colorectal cancer.

## 2. Materials and Methods

### 2.1. Data Source and Literature Search Strategy

The review was carried out in accordance with the Preferred Reporting Items for Systematic Reviews and Meta-Analysis guidelines [31]. The search was conducted according to the Cochrane Handbook for systematic reviews [32]. We searched PubMed, Embase^®^, MEDLINE, Allied & Complementary Medicine^TM^, Medical Toxicology & Environmental Health, DH-DATA: Health Administration for articles published before 1 October 2021. The keywords were as follows and different combinations of the terms related to “neuropilin-2,” “neuropilin 2,” “NRP2,” “NRP-2”, “colorectal cancer,” and “colon cancer,” were used in the search. This strategy was applied to all retrieved studies. The last search was performed on the 1 October 2021.We then looked at the references from the retrieved papers to see if any additional publications could be relevant. We discovered 97 studies in the databases Embase^®^, MEDLINE, Allied & Complementary Medicine, Medical Toxicology & Environmental Health, DH-DATA: Health Administration, and 12 papers in PubMed. A priori defined review protocol was registered for this systematic review (PROSPERO CRD42021294138).

### 2.2. Study Selection

Two authors (C.L. and V.G.) began reading all the titles and abstracts of the retrieved studies for inclusion. If discrepancies appeared between the two reviewers (C.L. and V.G.), a third author (D.S.) reviewed the articles and resolved any conflicts.

### 2.3. Inclusion and Exclusion Criteria

The papers included in the review fulfilled the following criteria:randomized controlled trials, cross-sectional, case-control, and cohort studies;studies with adult patients (18 years old or above) diagnosed with colorectal cancer;studies with NRP-2 data in relation to colorectal cancer;only peer-reviewed studies published in English with full text available.

It was decided to remove all case reports, non-full text, reviews, and meta-analyses, including those presented as meeting abstracts, posters, and correspondence (high risk of bias). Papers with questionable relevant data that failed to study NRP-2 expression in CRC were also excluded. A total of 10 studies met these standards.

### 2.4. Data Extraction

The titles and abstracts were independently screened by two authors (C.L. and V.G.), all publications which were identified as a result of the search for inclusion were categorized as ‘retrieve’ (eligible or possibly eligible/unclear) or ‘do not retrieve’ (ineligible or potentially eligible/unclear). We obtained the full-text of chosen manuscripts, and the authors independently examined the full-text and identified articles for inclusion, as well as grounds for exclusion of ineligible articles, and noted those reasons. Any issues were settled via conversation. Independently, the literature search, selection of studies that met the inclusion criteria, and data extraction were all completed. If the investigators’ choices for the number of articles disagreed, an agreement was reached following discussion. A third reviewer (V.G.) settled the eligibility issues between the first two reviewers. Finally, the findings were reviewed and finalized for inclusion. We sought the following outcome data and entered them in an Excel worksheet as follows: Population, Samples properties, Cancer type, Cell lines, Techniques used, Vectors and Genes or NRP-2 expression.

### 2.5. Quality Assessment—QUADAS 2 Tool

QUADAS 2 tool was used to evaluate the risk of bias and applicability of primary accuracy of the included studies in a review. A web app designed Robvis tool was used in the current review for visualizing risk-of-bias assessments. The tool creates:“Traffic light” plots of the domain-level judgments for each result;Weighted bar plots of the distribution of risk-of-bias judgments within each bias domain.

Bias is assessed as a judgment (high, low, or unclear) for individual studies from seven domains: sequence generation, allocation concealment, blinding of participants and outcomes, missing data, selective reporting, and others.

## 3. Results

### 3.1. Overview of Selected Studies

One hundred nine articles were identified through an online database search. Ten articles were removed after duplicates. The remaining ninety-nine articles were evaluated in detail and eighty-nine articles were excluded with reasons. Finally, ten articles [7,17,33,34,35,36,37,38,39,40] using search databases and various inclusion criteria were retrieved as shown in Figure 3.

### 3.2. Characteristics of the Selected Studies

Ten studies using cell lines, formalin-fixed paraffin-embedded tumors, healthy human tissue samples, and mice models were included in our review according to the inclusion and exclusion criteria—Table 1. Nine articles [7,17,33,35,36,37,38,39,40] included studies on multiple human CRC cell lines, eight [17,33,34,36,37,38,39,40] involved malignant human samples, while four [7,33,38,40] used tumor tissue harvested from mice models. Various techniques were applied including immunohistochemistry, PCR, western blotting, luciferase reporter assay, tumorigenicity assay, ELISA to assess the expression of either NRP1, NRP2 & TGF-β1, Sema3F, VEGFR2, XIST, Integrin α9 & β1.

Moreover, each of the 10 publications included in this analysis has been assessed by Cochrane for its risk of bias, as shown in Figure 4. The majority of the research had a low risk of bias.

### 3.3. NRP-2 up and Downregulation in CRC

The current systematic review looked at the effect of NRP-2 on CRC development, and its synergistic consequences with XIST, miR-486-5p, suggesting that they might be potential targets for diagnosing and treating CRC. For miRNAs to restrict the translation process by blocking mRNA translation or triggering mRNA degradation via contributing to the creation of the RISC complex, the 700–1200 amino acid AGO protein was necessary. AGO protein involvement in XIST regulation has therefore been hypothesized, however, XIST precursor and AGO protein reciprocal effects have yet to be discovered [36]. To combat multicellular resistance, NRP-2 interfering RNA might be utilized. Compared to normal tissues and cells, NRP-2 expressions were considerably elevated in CRC tissues and cells, according to Liu et al., who included 317 subjects confirmed with CRC postoperatively who had not previously received radiation or chemotherapy (Table 1).

Furthermore, Liu and colleagues found a link between miR486-5p and NRP-2 [36]. This miRNA suppressed NRP-2 expression in CRC cells, affecting apoptosis, proliferation, and the epithelial-mesenchymal transition. In addition to studying the interaction between NRP-2 and miRNA in CRC cells, Liu et al. discovered that miR331-3p exhibited a negative association with NRP-2 expression in 54 patients’ tissues (Table 1). According to RT-qPCR studies, MiR-331-3p expression in CRC cells was much lower than in normal colonic cells [37]. NRP-2 mRNA levels in CRC tissues and cells were found to be significantly higher than the relevant controls in an RT-qPCR study. CRC patients with decreased miR-331-3p expression had malignant clinic pathological characteristics, according to a clinic pathologic analysis. NRP-2 was discovered to be a new miR 331-3p target, and knocking it out restored the inhibitor’s effects on cell invasion and migration to some extent. These findings suggested that miR 331-3p had tumor-suppressive effects in CRC by targeting NRP-2 and that miR 331-3p/NRP-2 could be used as a CRC therapeutic [37]. SW480 cell invasion and migration were hampered by overexpression of miR-331-3p, whereas reduced expression supported SW480 cell migration and invasion. MiR-331-3p mimics risen E-cadherin expression while decreasing Vimentin and N-cadherin expression significantly. Over-expression of miR-331-3p inhibits CRC cell migration and invasion by regulating NRP2 and EMT [37].

In the cell lines tested, silencing of NRP-2 partially reversed miR331p inhibitor-mediated stimulation of cell invasion and migration, demonstrating that NRP-2 silencing partially reverses miR331p inhibitor-mediated activities. Another miRNA, miR-486-5p, has been shown in vivo to decrease tumor formation and lymphangiogenesis by targeting NRP-2 in CRC [38]. Neuropilin-2 is a direct functional target of miR-486-5p in CRC cells, according to a luciferase reporter assay, and overexpression of miR-486-5p in CRC cells is inversely related to neuropilin-2 expression. Overexpression of miR-486-5p inhibited tumor formation and lymphangiogenesis in nude mice, but overexpression of neuropilin-2 reversed this effect in the same animals. MiR-486-5p expression in SW620 and HT-29 CTC cells was measured using qPCR, and when compared to NRP-2 expression in these cells transfected with nontarget control vectors, miR-486-5p expression was significantly reduced at both the mRNA and protein levels. Liu et al. identified that miR-486-5p might be a tumor suppressor in the context of our research [38].

CRC cells stimulate NRP-2 in lymphatic endothelial cells involved in tumor lymphangiogenesis via integrin91/FAK/Erk pathway independent VEGF-C/VEGFR3 signaling [39]. Activating NRP-2 in LECs improves their migratory, sprouting, and tubulogenic capacity in vitro through guiding cytoskeleton polarity reorganization. In xenografts made from SEMA3F knockdown CRC cells, NRP-2 is significantly activated in tumor-associated LECs, resulting in significantly enhanced tumor lymphangiogenesis. Furthermore, CRC cells activate NRP-2 in LECs via VEGF-C/VEGFR3 signaling, which is independent of the integrin91/FAK/Erk pathway, to promote tumor lymphangiogenesis. The NRP-2 ligand SEMA3F reduces cancer cell adhesion and migration by interacting with integrins v3 and v4 in the setting of tumor necrosis by alpha [39]. According to a recent study [40], NRP-2 promotes cell migration in the presence of conditioned media from other types of cancer cells, but not from colorectal cancer cells. However, the limited number of tissues tested -seven- must be considered. NRP-2, a VEGFR2 co-receptor, was favorably related with vascularity in PNET tissues but not with VEGFR2. NRP-2 increased the migration of human umbilical vein endothelial cells (HUVECs) grown in the presence of conditioned media PNET cells via a VEGF/VEGFR2-independent route. NRP-2 also boosted F-actin polymerization by activating the actin-binding protein cofilin. Cofilin phosphatase slingshot-1 (SSH1) was abundantly produced by NRP-2-activating cofilin, and decreasing SSH1 enhanced NRP-2-activated HUVEC migration and F-actin polymerization. Furthermore, inhibiting NRP-2 in vivo decreased PNET angiogenesis and tumor development. Finally, higher NRP-2 expression has been associated with a poorer prognosis in PNET patients [33].

Staton et al. [34] revealed that although epithelial expression of both NRP-1 and NRP-2 increased considerably across the adenoma–carcinoma sequence—ACS (*p* = 0.0007, respectively *p* < 0.002), only NRP-1 was associated with microvessel density—MVD (*p* = 0.0003), and was moderately connected with VEGF (*p* = 0.001). Furthermore, while NRP1 vascular expression increased significantly across the ACS (*p* = 0.0004) and was positively correlated with MVD (*p* = 0.0006), NRP2 vascular expression decreased significantly (*p* = 0.0001) and was negatively correlated with MVD (*p* = 0.0001). As a result, neuropilins may play diverse functions in the angiogenic pathway during colorectal cancer development.

Grandclement et al. [7] findings suggest NRP-2 involvement in EMT after inducing cancer cells a fibroblastic form. The loss of epithelial markers like cytokeratin-20 and E-cadherin and the gain of mesenchymal markers like vimentin were linked with neuropilin-2 expression in CRC cell lines. Furthermore, NRP-2 promotes TGF-b1 signaling, resulting in constitutive phosphorylation of the Smad2/3 complex, and increased E-cadherin levels were seen after therapy with specific TGFb-type1 receptor kinase inhibitors.

Zheng et al. [35] demonstrated that Sema3F may sensitize multicellular resistance (MCR) to 5-fluorouracil and oxaliplatin through the use of the NRP-2 receptor by reducing integrin v3 expression.

Gray et al. [17] revealed that the expression of NRP-2 in CRC tumors was much greater than that in surrounding mucosa. Cell lines with reduced NRP-2 levels had reduced VEGFR1 signaling but unaffected cell growth. The study shows a reduction in the size of hepatic colorectal tumors in mice by a statistically significant margin (*p* = 0.005) compared to the control group, as a result of the in vivo targeting of NRP-2 by small interfering RNA (siRNA).

### 3.4. Bias Assessment and Applicability Judgements

QUADAS-II tool was used to individually assess the risk of bias and applicability judgments. The following listed items were assessed as being high, unclear, or low concerning bias: patient selection; index test; reference standard; flow and timing. Two reviewers evaluated the quality of selected studies and our assessment revealed that most of the included studies received a low or unclear risk of bias. The risk of bias graph and summary is shown in Figure 3. The overall results of the Cochrane assessment indicated that most of the studies (6) had a low risk of bias, except the study of Liu et al. [20] and Lou et al. [24] which had a high risk of bias. Beside these, two studies showed unclear risk assessment.

## 4. Discussion

In this systematic review, we analyzed ten studies that included 1118 patients with the precise aim of summarizing the existing literature on the role of NRP-2 in the EMT of colorectal cancer. To our knowledge, this is the first systematic review on this topic and was done to provide clinicians and researchers with a clear picture of this aspect.

The presence of the specific glycoprotein NRP-2 in neoplastic cell lines membranes and the fact that it was not detectable in nonmalignant colonic mucosa is highlighted by multiple studies [17,21,35,36,38]. Depending on the tumoral tissue, NRP-2 can be overexpressed in CRC [7,17,36] and breast cancer [21] or can be completely absent in cancers such as prostate cancer and B cell lymphoma [7].

According to immunofluorescence examination of several cancer cell lines and tumoral tissues, NRP-2 is expressed on the multiple human CRC cell lines membranes, but not in normal tissues [7]. When neuropilin-2 was found in colorectal cancer cell lines, researchers discovered that epithelial markers like cytokeratin-20 and E-cadherin were lost and mesenchymal molecules like vimentin were acquired.

Neuropilin-2 expression on colon cancer cell lines has been shown to enhance transforming growth factor-1 signaling, likely to result in Smad2/3 complex phosphorylation that persists. Neuropilin-2 and TGF-1 signaling work together to promote cancer development, and NRP-2 is involved in the transition from epithelial to mesenchymal cells [7].

The change from the epithelial to the mesenchymal phenotype is thought to prepare the way for a more aggressive kind of neoplasia [34], and tumors that express NRP-2 have a greater probability of becoming metastatic.

There is a link between the development of colorectal cancer and an increase in the expression of the NRP genes NRP-1 and NRP-2, as well as NRP-1 in the vascular cells. When it comes to adenomas and tumors, Staton’s research found that 73% and 88% of them resided in the left-side colon and rectum. NRP-1 can be identified in a single molecule in the mucosa of 40% of the colon. In dysplastic adenomas and colorectal cancers, the expression of epithelial cells increased significantly (*p* = 0.0007) and was no longer restricted to single cells [34]. Neuropilins, both dysplastic and cancerous, increased in expression throughout the ACS (adenoma–carcinoma sequence), suggesting a role in tumor growth.

Moreover, neuropilin expression was enhanced in both dysplastic and malignant tissues across the ACS, implying a role in tumor development [34]. Research shows a poorer prognosis for patients with coexpression of NRP-1 and NRP-2. Clinical development was not applicable to the patients with NRP-2 overexpression according to Staton et al. [34]. This assertion must be taken into consideration with moderation because it did not pass the threshold of statistical significance.

As a result of a high level of NRP-2 expression in all carcinomas, no correlation was established between MVD, VEGF, or NRP1. According to Gray et al. in 2008 [17], NRP-2 expression was much higher in tumors than in nonmalignant surrounding mucosa. Patients with colorectal cancer (CRC) who express both NRP1 and NRP-2 have a worse prognosis than those who express either one neuropilin or none at all [34]. The augmentation of NRP-2 receptor expression is closely connected to the increased expression of Sema3F and the reduced expression of integrin alphaVbeta3 [35].

On the other hand, the study by Lou et al. [40] that found NRP-2 is responsible for cell migration in pancreatic neoplastic cells when conditioned media is present, but not in lung or colorectal cancer cells, should be viewed with caution due to the small sample size (*n* = 7).

Additionally, unlike NRP-1, there are no published clinical trials concerning the possibility of targeting NRP-2 as a therapeutic method. A preliminary phase I research of a monoclonal antibody (MNRP1895A) directed against the VEGF binding region of NRP1 appeared promising and had an acceptable safety profile [41]. Yet, the subsequent phase Ib trial on the concurrent suppression of NRP1 and bevacizumab in combination with chemotherapy revealed a significantly greater rate of clinically significant proteinuria than expected, which does not warrant future testing of MNRP1685A in conjunction with bevacizumab [42].

Most studies -six- scored a low risk of bias, according to Cochrane assessment, and this is a strong point for the validity of our systematic review. Another considerable characteristic of all studies except Staton et al. is that cell lines were used, a helpful tool providing a consistent sample and reproducible results.

The strengths of this study are consisting of the comprehensive search strategy and the rigorous analytic framework that was applied. Following a systematized workflow granted by PRISMA, only 10 studies out of the initially 109 were taken into consideration in this systematic review, after following the three major steps: identification, screening, and final studies included.

Another strong point of this study consists in using the QUADAS-II tool, an effective tool used for quality assessment in studies with heterogenous design and conduct.

Furthermore, to our knowledge, this is the first systematic review that aims to clarify the role of NRP-2 in the EMT in CRC cells. This objective is of major importance considering the possibilities that NRP-2 to become a potential therapeutic target in CRC or other cancers where it is expressed.

The main limitations related to this systematic review are due to the rigorous inclusion criteria to ensure the high accuracy of the contents, all the more only a few data are available at this moment in the scientific literature about this subject. Inclusion and exclusion criteria may also restrict the extension of the results and although the comprehensive search was done only in well-established databases and extensive cross-referencing, we cannot exclude the possibility of having missed potentially relevant studies, in particular, if they were reported in languages other than English or grey literature.

The relatively small sample size and the fact that not all studies had gender info available are to be taken into consideration when conclusions are drawn.

Liu A. et al. state as a limitation that almost no animal models were used to confirm the fundamental relationship between miR-486-5p and NRP-2. Also, Gray et al. consider a limitation the usage of mostly HCT-116 human colorectal carcinoma cells.

## 5. Conclusions

The ten studies have found that NRP-2 is a promising therapeutic target since its expression is required for colorectal cancer cell proliferation and metastasis. NRP-2 expression was shown to be higher in the majority of the human primary and metastatic colon cancer cell lines when compared to the normal colonic mucosa. NRPs have been found in various human malignancies in addition to their expression in neovessels. The widespread expression of NRPs in many human malignancies suggests that this molecular network may play a role in cancer development. Although the NRP-1 importance in cancer has been well-known, the precise role of NRP-2 in oncogenesis has just lately been investigated. NRP-2 is expressed in human colon cancer cells but not in unassociated nearby mucosa, according to early studies. NRP-2 expression has been associated with the development of malignant phenotypes in colon, pancreatic, and breast malignancies. The NRP-2 is a possible therapeutic target in human malignancies where it is expressed and is critical for tumor formation in colorectal carcinoma cells.

## Figures and Tables

**Figure 1 biomedicines-10-00172-f001:**
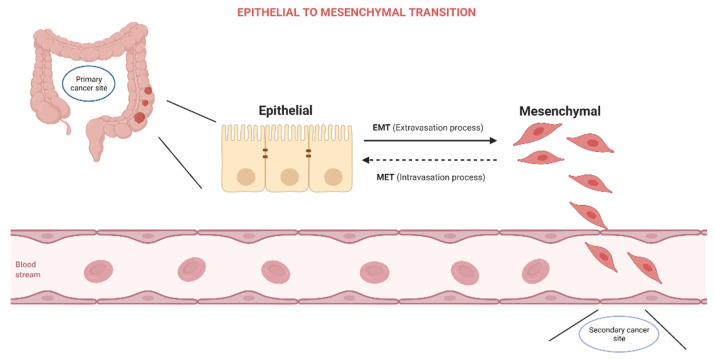
Epithelial to Mesenchymal Transition.

**Figure 2 biomedicines-10-00172-f002:**
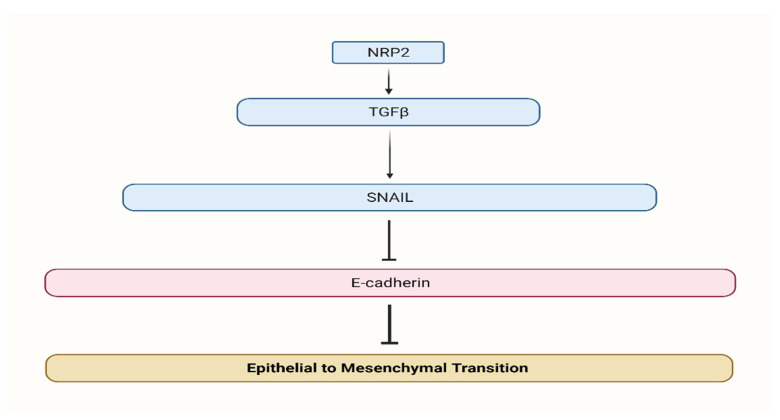
Neuropilin 2 regulatory mechanism. NRP2 promotes TGFβ signaling and induces EMT.

**Figure 3 biomedicines-10-00172-f003:**
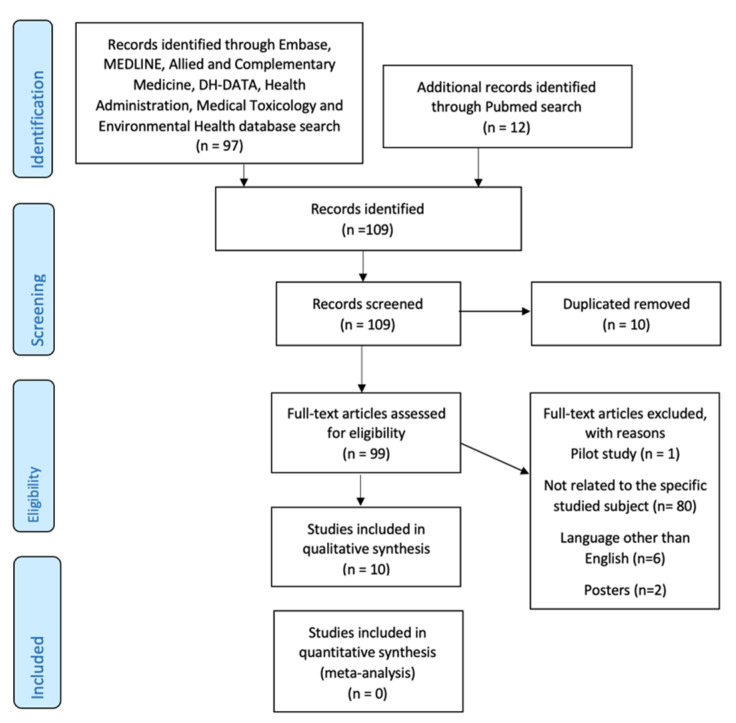
PRISMA flow diagram of the study selection process.

**Figure 4 biomedicines-10-00172-f004:**
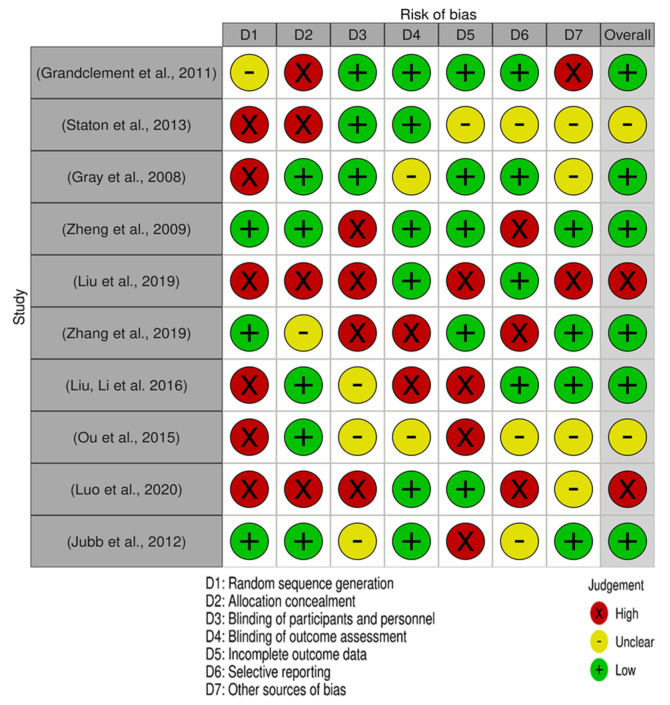
Risk of bias summary for each included study. The Cochrane risk-of-bias assessment for the 10 articles included in this review. In terms of random sequence generation, 6 studies had a low risk of bias while 4 had high risk of bias. Concerning the risk of allocation concealment, 5 studies had a low risk of bias and 5 has a high risk of bias. In the selective reporting domain, 4 studies had a low risk of bias, 3 unclear, and 3 had high.

**Table 1 biomedicines-10-00172-t001:** Characteristics and summary of included studies.

No. crt.	Author	Population	Sample Properties	Cancer Type	Cell Lines	Assays/Techniques Applied			Vectors	Plasmid	Gene/NRP-2 Expression	NRP-2 Correlation with CRC		
1	Grandclement C. et. al. [7]	11	Tumor cell lines	Colorectal cancer cells	HT29, Colo320, SW620, MCF7	STR analysis, Flow cytometry, Cell	P proliferation and ELISA assay		hNRP-2 (pcDNA3.1-NRP-2 &	pCMV6-XL5-NRP-2	NRP-2 & TGF-β1expressed	NRP-2 role in epithelial-mesenchymal transition is suggested. The cross-talk between NRP-2 and TGF-b1	signaling supports cancer progression.	NRP-2 is expressed at the membrane of several human CRC cell lines, while it is not expressed in normal tissues.
2	Staton CA et al. [33]	167	Aged persons	Epithelial and colorectal	-	Immunohistochemistry			-		NRP-1 and NRP-2 increased	NRP-2 is overexpressed in all cancers. No relevant connection with MVD, VEGF, or NRP-1 was discovered.	Poor prognosis was more likely for the patients that expressed NRP-1 and NRP-2, compared with	the ones expressing only NRP-1 or NRP-2 or even no NRP at all.
3	Gray MJ et al. [17]	10	Derivation from sh-Con	Colon adenocarcinoma	HCT-116, HT-29, RKO, SW480 & KM12	Immunoprecipitation & Immunoblot analysis RT-PCR, (ELIA) and MTT analysis			shRNA		NRP-2 expressed	Tumors presented a	NRP-2 was overexpressed in tumors compared with the adjacent mucosa	
4	Zheng C et al. [34]	5	After chemotherapy	Ovarian cells	HT29, HCT116, and Lobo	3D cell culture, SEM and TEM analysis, Western blotting & Chemosensitivity			Sema3F cDNA pSecTag		Sema3F & NRP-2 increase	The up-regulation of NRP-2 receptor expression was linked to elevated expression of Sema3F and down-regulation of integrin alphaVbeta3 expression		
5	Liu A et al. [35]	317	CRC Population	Rectum & Colon	HCT116, HT29 & SW620	RT-PCR, Western blotting, Cell apoptosis, and cell invasion assay			IncRNA3.1-XIST		XIST & NRP-2 high	miR-486-5p is decreased in CRC tissues. By directly targeting NRP-2, it attenuated in vivo the proliferation and lymphangiogenesis of the tumoral cells		
6	Zhang H et al. [36]	54	CRC Population		Colorectal carcinoma Colon and rectum	SW480 & HCT116)	RT-qPCR, Transwell assays, western blot analysis, and IHC		pGL3	luciferase vectors (Promega)	miR-331-3p expressed	NRP-2 was upregulated in CRC		
7	Liu C et al. [37]	66	Before chemotherapy		Malignant gastrointestinal tumor	SW620 and HT-29	qRT-PCR, Western blot, luciferase reporter assay, Tumorigenicity assay, and in silico analysis		pMIR-GLO		miR-486-5p downregulated and NRP-2 was overexpressed.	Normal tissues and cells had lower NRP-2 expression by comparison with CRC tissues and cells		
8	Ou J-J et al. [38]	200	After surgery		Tumor lymphatic endothelial cells	SW480 & SW620	ELISA, qPCR & Transwell assay &	Tubologenisis assay	FAK shRNA & Rac1 siRNA & RNAi (TG320362)		Integrin α9 & β1, XIST & NRP-2 low expressed	The density of tumor lymphatic vessels is linked to NRP-2 expression levels in CRC		
9	Luo X et al. [39]	13	Non-small-cell lung &	colon cancer	Pancreatic neuroendocrine tumors (pnets	HUVECs & BON	Western blotting, Immunohistochemistry, Cellular F-actin/G-actin assay, Phalloidin staining say		pGC-LV-GV308		VEGFR2	If conditioned medium from colorectal is present, NRP-2 doesn’t act to promote cell migration		
10	Jubb AM et [32]	275	Paraffin-embedded tissue		Lung, breast, and Colo-rectal cancer	NSCLC lines	Immunohistochemistry		-		Neuropilin-2 expressed	Blood and/or lymphatic vessels showed immunoreactivity for NRP-2.	NRP-2	protein expression was detected in tumor cells from 22% of colorectal cancers

## Data Availability

The data that support the findings of this study are openly available.

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
