# Peer review of "The Role of Neuropilin-2 in the Epithelial to Mesenchymal Transition of Colorectal Cancer: A Systematic Review"

_biomedicines, 2022, doi:10.3390/biomedicines10010172_

Round 1
Reviewer 1 Report
The manuscript mainly focuses on the roles of NRP-2 in the process of CRC EMT. However, NRP-1 is also mentioned in both the Results and Discussion sections. More importantly, one of the studies indicated that both NRP-2 and NRP-1 are required for tumor progression, and the overexpression of NRP-2 itself was not enough to promote tumor growth. Thus, the authors should provide more introduction about the difference between NRP-1 and NRP-2 with regard to their functions in tumor development. They should also discuss the possibility of targeting NRP-1 and NRP-2 in clinic for CRC treatment. Any previous efforts, either in pre-clinical studies or clinical trials, on targeting NRP-2 for CRC treatment should also be discussed and cited in the manuscript.
Overall, the manuscript is well-organized and the English language is OK. But some grammatical mistakes and typos can be seen, e.g.: line38: "EMT and is classified into three types"; line133: "one hundred nine articles.."; line163: "and it's synergistic consequences...". The authors may want to perform a careful proofreading of their manuscript to eliminate any typos/grammatical mistakes like these.
Author Response
We want to thank the reviewer for taking the time to revise our manuscript. We are confident that making the suggested modifications we will improve our manuscript.
We thank the reviewer for the on-point comment. Considering the existing literature, there is scarce evidence about NRP-1 and NRP-2. We mention that the main focus of this systematic review is limited to NRP-2 in CRC. We promise that we will systematically search the existing databases for a future NRP-1 systematic review /meta-analysis to close the existing literature gap. For the moment is out of the scope of the review. We have amended our manuscript so that the readers get an insight about the role of NRP-1. We added a new paragraph in the introduction and also in the discussions section:
”Either NRP-1 or NRP-2 or both are expressed in nearly all tumor cells. Tumor development, progression, and therapy options have all been examined for NRP-1 [30]. Solid tumor development is aided when NRP-1 binds VEGF-A, while SEMA3A binding often improves prognosis by limiting tumor cell motility and invasion. Furthermore, via activating TGF-β pathway, NRP-1 promotes EMT and contributes to metastasis.”
We have added the following paragraph to the discussions to highlight the existing clinical trials and possible therapeutic measures:
Additionally, unlike NRP-1, there are no published clinical trials concerning the possibility of targeting NRP-2 as a therapeutic method. A preliminary phase I research of a monoclonal antibody (MNRP1895A) directed against the VEGF binding region of NRP1 appeared promising and had an acceptable safety profile [41]. Yet, the subsequent phase Ib trial on the concurrent suppression of NRP1 and bevacizumab in combination with chemotherapy revealed a significantly greater rate of clinically significant proteinuria than expected, which does not warrant future testing of MNRP1685A in conjunction with bevacizumab [42].
As we are not native English speakers, our Professional English Editing team has made a second verification of the punctuation, grammar, syntax and spelling of our article.

Reviewer 2 Report
This is a good literature review (according to the Cochrane guidelines) on the expression of neuropilin-2 and its role in the epithelial to mesenchymal transition in solid tumors, particularly in colorectal cancer.
Comments:
The authors should state in the introduction why they focused only on NRP-2 and did not include NRP-1 in their review. NRP-1 could also be a potential target in solid tumors including CRC…
A scheme or graphical abstract illustrating and summarizing the regulatory mechanisms of NRP-2 expression in CRC, NRP2 / TGFb interactions, signaling pathways etc, should be added.
Author Response
We want to thank the reviewer for taking the time to revise our manuscript and for the kind words. We are confident that making the suggested modifications we will improve our manuscript.
We thank the reviewer for the comment. The main focus of this systematic review is NRP-2 in CRC. We promise that we will systematically search the existing databases for a future NRP-1 systematic review /meta-analysis to close the existing literature gap. For the moment is out of the scope of the review. We have amended our manuscript for the readers to get an insights about the role of NRP-1. We added a paragraph in the introduction section. Now the text can be read as:
”Either NRP-1 or NRP-2 or both are expressed in nearly all tumor cells. Tumor development, progression, and therapy options have all been well examined for NRP-1 [30]. Solid tumor development is aided when NRP-1 binds VEGF-A, while SEMA3A binding often improves prognosis by limiting tumor cell motility and invasion. Furthermore, via activating TGF-β pathway, NRP-1 promotes EMT and contributes to metastasis.”
We thank the reviewer for the suggestion. We have amended our manuscript to include a graphic scheme about the regulatory mechanisms of NRP-2 expression in CRC.
